# New Transition Metal Coordination Polymers Derived from 2-(3,5-Dicarboxyphenyl)-6-carboxybenzimidazole as Photocatalysts for Dye and Antibiotic Decomposition

**DOI:** 10.3390/molecules28217318

**Published:** 2023-10-28

**Authors:** Yu Wu, Wenxu Zhong, Xin Wang, Weiping Wu, Mohd. Muddassir, Omoding Daniel, Madhav Raj Jayswal, Om Prakash, Zhong Dai, Aiqing Ma, Ying Pan

**Affiliations:** 1School of Chemistry and Environmental Engineering, Sichuan University of Science & Engineering, Zigong 643000, China; 2Department of Chemistry, College of Sciences, King Saud University, Riyadh 11451, Saudi Arabia; muddassir@ksu.edu.sa; 3Department of Chemistry, Faculty of Science, University of Lucknow, Lucknow 226007, India; omodanzo@gmail.com (O.D.); madhavrajjayswal@gmail.com (M.R.J.); 4Guangdong Provincial Key Laboratory of Research and Development of Natural Drugs, and School of Pharmacy, Guangdong Medical University, Guangdong Medical University Key Laboratory of Research and Development of New Medical Materials, Dongguan 523808, China

**Keywords:** coordination polymers, antibiotics, dyes, photocatalysis, Hirshfeld surface

## Abstract

Coordination polymers (CPs) are an assorted class of coordination complexes that are gaining attention for the safe and sustainable removal of organic dyes from wastewater discharge by either adsorption or photocatalytic degradation. Herein, three different coordination polymers with compositions [Ni(HL)(H_2_O)_2_·1.9H_2_O] (**1**), [Mn_3_(HL)(L)(μ_3_-OH)(H_2_O)(phen)_2_·2H_2_O] (**2**), and [Cd(HL)_4_(H_2_O)]·H_2_O (**3**) (H_3_L = 2-(3,5-dicarboxyphenyl)-6-carboxybenzimidazole; phen = 1,10-phenanthroline) have been synthesized and characterized spectroscopically and by single crystal X-ray diffraction. Single crystal X-ray diffraction results indicated that **1** forms a 2D layer-like framework, while **2** exhibits a 3-connected net with the Schläfli symbol of (4^4^.6), and **3** displays a 3D supramolecular network in which two adjacent 2D layers are held by π···π interactions. All three compounds have been used as photocatalysts to catalyze the photodegradation of antibiotic dinitrozole (DTZ) and rhodamine B (RhB). The photocatalytic results suggested that the Mn-based CP **2** exhibited better photodecomposition of DTZ (91.1%) and RhB (95.0%) than the other two CPs in the time span of 45 min. The observed photocatalytic mechanisms have been addressed using Hirshfeld surface analyses.

## 1. Introduction

Release of a significant quantity of organic compounds, especially aromatic dyes and antibiotics, in water bodies from industries and residential areas has resulted in considerable water pollution, which not only destroys the aquatic environment but also poses safety problems for humans and animals [1,2,3,4,5,6,7,8,9,10]. Thus, developing advanced methods and materials to safely eliminate these classes of compounds from wastewater must be taken into consideration by chemists and environmentalists. Coordination polymers (CPs) are an assorted class of coordination complexes that are designed and synthesized by coordinating rationally selected polydentate ligands with targeted metal cation centers. Such strategy gives rise to varied types of two-dimensional and three-dimensional frameworks in this class of crystalline materials that find applications as adsorbents [11,12,13,14,15,16,17,18,19], as supercapacitors [20,21], in drug delivery [22,23,24,25,26,27,28,29,30,31], as sensors [32,33,34,35,36,37,38,39,40,41,42], and in catalysis [43,44,45,46] and drug delivery [47,48]. Because of fascinating characteristics such as peculiar architectures and homogenous distribution of active sites, this type of material has received significant attention as photocatalytic materials [49,50,51,52,53]. Wang et al., for example, reported Fe-based MOFs (Fe-MIL-101) for tetracycline (TC) removal under visible light illumination, with a TC removal rate of 57.4% [54,55].

The selection of suitable ligands is critical in order to construct tailored coordination polymers. Among the many classes of ligands, aromatic multicarboxylic acids have been extensively employed as building blocks for designing and fabricating the desired CPs [56,57,58,59,60,61,62]. Among the widely used group of multicarboxylic acid ligands, the rigid blocks having at least two aromatic rings and several COOH groups arranged in different positions are capable of forming fascinating structures. Moreover, such ligands induce functionally active characteristics in the CPs [63,64,65,66]. Additionally, the use of ancillary ligands (heterocyclic N- or N,N-donor) can also be employed to stabilize and extend the structures for designing newer frameworks along with these multicarboxylate building blocks [67,68,69,70,71,72].

These characteristics of the multicarboxylate ligands therefore motivated us to investigate the design and synthesis of new CP-based photocatalysts. Hence, in the presented investigation, three new transition metal-based CPs have been synthesized using 2-(3,5-dicarboxyphenyl)-6-carboxybenzimidazole as the main ligand. This ligand was selected due to its semi-rigid nature and the presence of three carboxyl groups, which can generate multiple nodes and enhance thermal stability and overall rigidity of the frameworks. In view of this, three new Ni(II)/Mn(II)/Cd(II)-CPs with the aforementioned ligand have been synthesized, characterized, and used as photocatalytic materials to catalyze the photodegradation of dinitrazole antibiotic and rhodamine B dyes. The outcomes of these studies are presented herewith.

## 2. Results and Discussion

### 2.1. Structural Description of **1**[Ni(HL)(H_2_O)_2_·1.9H_2_O]

The single crystal X-ray diffraction analysis revealed that CP **1** crystallizes in the monoclinic unit cell with a *C*2/c space group. The asymmetric unit of **1** consists of one Ni(II), one HL^2−^ ligand, two coordinated water molecules, and 1.9 free water molecules (Figure 1a). The immediate geometry around the Ni(II) is distorted octahedral, wherein the equatorial plane is occupied by four O atoms (O1, O3, O5 and O6) from three HL^2−^ ligands (Ni1-O1 = 1.980(3) Å; Ni1-O3 = 1.981(3) Å; Ni1-O5 = 2.126(3) Å; and Ni1-O6 = 2.103(3) Å), while the axial positions are occupied by two O atoms (O7, O8) of aqua ligands that are transverse with respect to each other (Ni1-O7 = 2.090(4) Å and Ni1-O8 = 2.059(4) Å) with ∠O7-Ni1-O8 of 175.12(14)° (Appendix A). Interestingly, the two HL^2−^ ligands exhibit monodentate coordination and the third ligand displays bidentate chelating coordination with the Ni(II) center. In CP **1**, each HL^2−^ connects three Ni^2+^ centers with a linkage angle of 5.1°; in addition, the two aromatic rings are almost parallel, with a dihedral angle of 7.2°. The three carboxylic groups in HL^2−^ show (κ^1^ − κ^0^)-*μ*_1_–COO^−^ monodentate mode and (κ^1^ − κ^1^)-*μ*_1_–COO^−^ chelating mode, and one remains as –COOH. These coordination modes of HL^2−^ linkers join the octahedral Ni(II) to form a 2D layer (Figure 1b). The framework robustness in the 2D layer is provided by four types of weak interactions (Figure 1d): (i) the aqua ligand hydrogen (H7B) forms the intermolecular hydrogen bond with neighboring uncoordinated carboxylate oxygen (O2), with a O7-H7B···O2 interaction distance of 2.21 Å; (ii) the carboxylate oxygen (O6) hydrogen bonds with the H7A hydrogen of the coordinated aqua ligand O7 of the adjacent molecule with a O6···H7A interaction distance of 1.97 Å; (iii) the coordinated water molecule (O8) hydrogen H8B forms hydrogen bonding with the carboxylic O atom (O5) of the neighboring moiety with O8-H8B···O5 separation of 2.14 Å; and (iv) the coordinated water molecule (O8) hydrogen (H8A) forms another hydrogen bond with the adjacent nitrogen N1 of the imidazole ring, having O8-H8A···N1 interaction distance of 2.06 Å (Appendix A). Additionally, two 2D layers are connected by the hydrogen bonds in face-to-face mode to engender a 3D supramolecular network using π···π interactions between the HL^2−^ ligand and two adjacent 2D layers (Figure 1e). From a topological point of view, each HL^2−^ acts as a 3-connected node, and hence each Ni^2+^ center can be regarded as a 3-connected node. Thus, the resulting structure of CP **1** is a 3-connected hexagonal-type topology with a Schläfli symbol of (6^3^) (Appendix A).

### 2.2. Structural Description of ***2***

The single crystal X-ray analysis reveals that CP **2** crystallizes in the triclinic unit cell with a *P*-1 space group. The asymmetric unit of **2** possesses three Mn(II), one HL^2−^ ligand, one L^3−^, one coordinated aqua ligand, and one hydroxyl group along with two chelating phen ligands and two free water molecules (Figure 2a). The immediate geometry around Mn1 is distorted octahedral with four oxygen center atoms coming from carboxylate ligands HL^2−^ and L^3−^, one bridging O of the OH group and one aqua ligand. The O1, O4, O9, and O11 are located at the equatorial plain, while the O13 and O14 oxygen atoms occupy the axial positions (Mn1-O9 = 2.303(4) (3) Å; Mn1-O11 = 2.164(4) Å) (Appendix A). In addition, the bond angles around Mn1 vary between 81.87(16)°–178.14(15)°, and the distortion around Mn1 can be ascribed to the Jahn-Teller effect. Apart from this, Mn2 and Mn3 display similar trigonal bipyramidal geometry (maximum atomic deviation of 0.018 Å), wherein Mn2 and Mn3 are displaced by 0.251 Å from the trigonal plane toward the strongly coordinated axial ligand. Moreover, **2**, the carboxylate groups of HL^2−^, adopt two different coordination modes *viz.* (κ^1^ − κ^1^)-*μ*_2_-COO^−^ and (κ^1^ − κ^0^)-*μ*_1_-COO^−^, respectively. The two differently coordinated HL^2−^/L^3−^ ligands link Mn(II) centers to form a 2D layer along the *ab* plane (Figure 2b), but this 2D layer framework is different from **1**. The phen ligands protrude from the 2D layer and are distributed regularly on the same side of the layer (Figure 2c). Furthermore, the phen and HL^2−^ ligands form weak π···π interactions with a separation of 3.741 Å (Figure 2d). From a topological point of view, each HL^2−^ acts as a 3-connected node, and hence each Mn^2+^ center can be regarded as a 3-connected node. Thus, the resulting structure of CP **2** is a 3-connected net with a Schläfli symbol of (4^4^.6) (Appendix A).

### 2.3. Structural Description of ***3***

The CP **3** crystallizes in the monoclinic unit cell with the *C*2/c space group, and the asymmetric unit consists of one independent Cd(II) ion, one HL^2-^ ligand, one aqua ligand, and one free water molecule (Figure 3a). The geometry around Cd(II) is distorted octahedral, wherein the equatorial plane is occupied by four O atoms (O1, O2, O5, and O6) from three HL^2−^ ligands, in which one HL^2−^ ligand coordinates in bidentate chelating mode and the other two in monodentate bridging mode between two Cd(II) centers, while the two trans axial positions are adopted by two O atoms (O3, O7) from the carboxylate oxygen of HL^2−^ and aqua ligands, respectively (Cd1-O3 = 2.367(8) Å and Cd1-O7 = 2.337(8) Å) (Appendix A). The ∠O3-Cd1-O7 is 175.7(3)° (Appendix A), indicating that Cd1 lies at the equatorial plane of the octahedron. In CP **3**, each HL^2−^ connects four Cd(II) centers with a bond angle of 2.9°, and the two aromatic rings are almost parallel, with a dihedral angle of 3.1°. The carboxylic groups in HL^2−^ exhibit (κ^1^ − κ^0^)-*μ*_1_–COO^−^ monodentate mode, (κ^1^ − κ^1^)-*μ*_1_–COO^−^ chelating mode, and (κ^1^ − κ^1^)-*μ*_2_–COO^−^ bridging mode. These coordination modes of HL^2−^ linkers join the octahedral Cd(II) to produce a layer (Figure 3b). The 2D layer is held by bifurcated hydrogen-binding interactions (Figure 3d): (i) The coordinated aqua ligand (O7) forms the intermolecular hydrogen bond to the neighboring uncoordinated carboxylic O atom (O4) with O7-H7A···O4 separation of 1.90 Å. (ii) The same aqua ligand is involved in hydrogen bonding with another carboxylate oxygen atom (O6) with O7-H7B···O6 distance of 2.07 Å (Appendix A). Further, the two face-to-face 2D layers are connected by the hydrogen bonds to generate a 3D supramolecular network (Figure 3c). Additionally, the π···π interaction distance between the HL^2−^ ligands is 3.589 Å, and two adjacent 2D layers propagated into a 3D supramolecular network (Figure 3e). From a topological point of view, each HL^2−^ acts as a 3-connected node, and hence each Cd^2+^ center can be regarded as a 6-connected node. Thus, the resulting structure of CP **3** is a 3,6-connected net with a Schläfli symbol of {4^2^.6}_2_{4^4^.6^9^.8^2^} (Appendix A).

### 2.4. FTIR Spectroscopy, Thermogravimetric Analyses, PXRD, and BET Surface Area Analysis

In the FTIR spectra of all three CPs, the relatively weak band appearing at ~3000 cm^−1^ is attributed to ν_C-H_ of the aromatic ring. The broad absorption bands in the range of 3400–3200 cm^−1^ may be assigned to the characteristic peaks of O–H stretching vibrations from coordinated or lattice water molecules. The appearance of a band ~1700 cm^−1^ suggested partial deprotonation of H_3_L ligands. In addition, bands between 1550–1600 cm^−1^ and 1340–1380 cm^−1^ can be assigned to ν_asymm_ and ν_symm_, respectively, of the COO^−^ groups of the coordinated ligand (Appendix A).

Thermogravimetric studies (TGA) were also carried out to learn more about the thermal stabilities of all three CPs. CP **1** shows two distinct weight loss regions in the TGA curve. The first sharp weight loss occurred at ca. 238 °C due to the loss of aqua ligands (obsd. 8.2%, calcd. 8.6%). Furthermore, the TGA of CP **1** revealed weight loss occurring between 210 and 755 °C, corresponding to the decomposition of the organic moiety with the formation of Ni_2_O_3_ (obsd. 37.9%, calcd. 39.4%). In CP **2**, the first sharp weight loss occurred until ca. 194 °C due to the release of the coordinated aqua ligand and co-crystallized water molecules (obsd. 5.3%, calcd. 5.7%). Moreover, it is worthy to mention here that the co-crystallized water molecules were release from the lattice of **2** at ca. 89 °C. Further, weight loss between 390–855 °C corresponded to the decomposition of the organic group to engender Mn_2_O_3_ (obsd. 39.8%, calcd. 37.9%). CP **3** exhibits two weight loss regions in its TGA curve. The first weight loss between 45–212 °C corresponds to the loss of the coordinated aqua ligand and co-crystallized water molecules (obsd. 7.9%, calcd. 7.6%). Similarly, in the second step, the framework of **3** collapsed with the loss of organic species quickly from temperature range 450–890 °C to form CdO (obsd. 26.9%, calcd. 27.1%). Additionally, powder X-ray diffraction (PXRD) experiments were conducted to evaluate the phase purity of all three CPs. The experimental PXRD profiles of all three CPs match satisfactorily with simulated PXRD profiles, indicating their bulk phase purity (Appendix A). Additionally, UV-Vis spectroscopy was used to evaluate the optical band gaps of MOFs that revealed absorption edges at 322 nm. On this basis, the optical band gap energies were calculated by the Kubelka-Munk function: αhν = A(hν − Eg)^n/2^, in which α stands for the absorption coefficient, hν stands for the photon energy, A is a constant, and Eg is the band gap. The band gaps for **1**, **2,** and **3** were calculated to be 2.10, 2.82, and 2.23 eV, respectively (Appendix A), indicating their semiconducting properties. Hence, CPs **1**–**3** can be utilized as photocatalysts for nitrophenol photodegradation.

The textural properties of **1**–**3** were explored using the BET N_2_ adsorption isotherm method performed at 77 K. It was observed that the isotherms displayed a continuous increase at low relative pressure (P/P_0_ < 0.05) and displayed a small hysteresis loop at high relative pressure (0.45 < P/P_0_ < 1), which indicated the existence of micropores in these CPs. In addition, CP **2** possesses relatively larger BET surface area and pore volume than CPs **1** and **3,** which might be arising because of the dense structural packing in later CPs. Furthermore, the DTZ pore size distribution in all the materials is smaller than 0.50 nm (Appendix A).

### 2.5. Photocatalytic Applications

After firm structural characterization of all three CPs, their applicability as photocatalysts were assessed by employing previously reported methods [73,74,75,76] against the decomposition of the dinitrazole (DTZ) antibiotic, which was assessed under UV light irradiation at a mean wavelength of 365 nm (Figure 4). Before performing the photocatalysis, attempts were made to establish the adsorption–desorption equilibrium by keeping the reaction mixture in the dark for 30 min and monitoring the UV-Vis spectra for all three reaction mixtures (Figure 4a–c) and also by constructing a graph of C/C_0_ vs. time plot (Figure 4d). Such investigation revealed little alternation in the absorption intensity of DTZ within this time span, thereby indicating the establishment of the adsorption–desorption between DTZ and CPs **1**–**3**. Further, the photocatalytic experiments indicated that the electronic absorption intensity of the DTZ solution decreased with time (Figure 4). In addition, in the timeframe of 45 min, the percentage decomposition of DTZ in the presence of photocatalysts **1**, **2,** and **3** was 84.2%, 91.1%, and 75.1%, respectively (Figure 4a–c). However, in the absence of these photocatalysts, the percentage photodegradation of DTZ was merely 18.2% (Figure 4d). Moreover, the percentage photodegradation of DTZ in the presence of the metal salts Ni(NO_3_)_2_·6H_2_O, Mn(NO_3_)_2_·4H_2_O, and Cd(ClO_4_)_2_·6H_2_O was 28.1%, 31.1%, and 25.3%, respectively (Appendix A). Hence, the photocatalytic performance of pristine metal salts revealed that the photocatalysis is not only arising because of the metal centers of the CPs 1–3; rather, the frameworks of CPs are collectively playing a role in their photocatalytic activities, photodegrading DTZ. Furthermore, the photodegradation of DTZ in the presence of these CP-based photocatalysts displayed pseudo-first-order kinetics (Figure 4e) and the rate constants for all three CPs calculated by using the plot of ln(*C_t_/C*_0_) vs. time (Table 1) suggested that the rate constant k (min^−1^) was highest in the presence of CP **2**. Hence, the percentage photodegradation of DTZ and the rate constant parameters further suggested that amongst all three CPs, the Mn(II) CP **2** displayed the best photocatalytic efficiency. These differences in the catalytic activities of the CPs might be arising because of the variation in the structural environments around metal centers [77,78,79].

Moreover, to assess the nature of possible active species generated in situ that are responsible for the photodecomposition of DTZ in the presence of photocatalyst **2,** radical scavenging experiments were performed. For this purpose, excess tertiary butyl alcohol (TBA) as the •OH scavenger, benzoquinone (BQ) as the superoxide (O_2_˙^−^) scavenger, and ammonium oxalate (AO) holes (h^+^) scavenger were added in three separate reaction mediums comprising DTZ and CP **2** under similar reaction conditions [80,81,82,83,84,85,86,87,88,89] (Figure 5). The experiments suggested that the order of photodecomposition as evident by the percentage decolorization of DTZ in the presence of different scavengers as well as CP **2** follows the order AO > TBA > BQ (Figure 5b,c). This was also evident by the significant decrease in the pseudo-first-order rate constant (k) from 0.0467 min^−1^ to 0.0097 min^−1^ when BQ was used as the scavenger (Figure 5c). This suggested that superoxide (O_2_˙^−^) is the main active species responsible for the decomposition of DTZ in the course of photocatalysis [90,91,92]. In addition, to appraise the stability and reusability of **2**, the photocatalytic performance of CP **2** was tested for its recycled sample (Figure 5d), which indicated no perceptible decline in the photocatalytic activity after four catalytic cycles.

Apart from DTZ, the photocatalytic performances of all three CPs were tested against the photodecomposition of rhodamine B (RhB). In this case also, before performing the photocatalysis, the adsorption–desorption equilibrium was established by keeping the reaction mixture in the dark for 30 min and monitoring the UV-Vis spectra for all three reaction mixtures (Figure 6a–c) and constructing a graph of C/C_0_ vs. time plot (Figure 6d). Such investigation revealed little alternation in the absorption intensity of EhB within this timespan, thereby indicating the establishment of the adsorption–desorption between RhB and CPs **1**–**3**. In these experiments too, the electronic absorption intensity of the RhB solution deteriorated with time (Figure 6) in the presence of all three photocatalysts **1**–**3**. Moreover, in the timespan of 45 min, the percentage decomposition of RhB in the presence of photocatalysts **1**, **2,** and **3** was 83.7%, 95.0%, and 81.1%, respectively (Figure 6a–c). However, without these photocatalysts, the percentage photodegradation of RhB under similar reaction conditions was merely 15.1%, which evinced that the CPs can photocatalyze the degradation of RhB as well. Moreover, the percentage photodegradation of RhB in the presence of the metal salts Ni(NO_3_)_2_·6H_2_O, Mn(NO_3_)_2_·4H_2_O, and Cd(ClO_4_)_2_·6H_2_O was 23.3%, 26.1%, and 28.4%, respectively (Appendix A). Hence, the photocatalytic performance pristine metal salts revealed that the photocatalysis is not only arising because of the metal centers of the CPs 1–3; rather, the frameworks of CPs are collectively playing a role in their photocatalytic activities, degrading RhB. In addition, like DTZ photodecomposition, the photodegradation of RhB also followed pseudo-first-order kinetics (Figure 6d), and the rate constants for all three CPs calculated by using the plot of ln(*C_t_/C*_0_) vs. time (Table 1) suggested that the rate constant k (min^−1^) for RhB degradation was highest in the presence of CP **2**. Hence, the percentage photodegradation of RhB and the rate constant parameters further suggested that amongst all three CPs, Mn(II) CP **2** displayed the best photocatalytic efficiency, which might be because of the variation in the structural environments around the metal centers [77,78,79].

Furthermore, for RhB photodegradation, **2** radical scavenging experiments in the presence of a photocatalyst were performed [80,81,82,83,84,85,86,87,88,89] (Figure 7). The experiments suggested that the order of photodecomposition as evident by the percentage decolorization of DTZ in the presence of different scavengers as well as CP **2** follows the order TBA > AO > BQ (Figure 7b,c). This was also made evident by the significant decrease in the pseudo-first-order rate constant (k) from 0.0669 min^−1^ to 0.0078 min^−1^ in the presence of BQ (Figure 7c). Hence, these scavenging experiments indicated that in this case also, the superoxide (O_2_˙^−^) radicals are the main active species responsible for RhB decomposition during photocatalysis [90,91,92]. Further, it was observed that similar to the photocatalysis of RhB, in this case also, the CP **2** indicated no perceptible decline in the photocatalytic activity after four catalytic cycles (Figure 7d).

Furthermore, the PXRD experiments performed for all three recovered CPs after photocatalysis after DTZ and RhB decomposition indicated no significant changes in 2θ positions when compared to the pristine batches of these CPs (Appendix A). However, slight broadening in the peaks in the recycled samples as compared to pristine batches may be attributed to a decrease in the particle size of the CPs during photocatalysis. This suggested good stability of all three photocatalysts during the photodegradation process and authenticated the structural framework integrity of CPs during the photocatalysis process.

Furthermore, to have information about the interaction between CPs and DTZ/RhB molecules, the molecular Hirshfeld surfaces in the crystal structures of **1**–**3** were constructed by using the procedure mentioned previously (Figure 8, Figure 9 and Figure 10) [93,94,95]. In all three CPs, the *d*_norm_ plots revealed prominent red circular depressions, indicating strong interaction zones, and the comparatively weaker interactions are represented by diffuse red areas (Figure 8a, Figure 9a, and Figure 10a). Additionally, the fingerprint plots (Figure 8d, Figure 9d, and Figure 10d) indicated that amongst all three CPs, CP **2** exhibits maximum O···H and N···H interactions. This indicates that **2** possesses the best capability to form weak interactions with DTZ/RhB molecules for their concomitant photodegradation. In addition, the crystal lattice void spaces for all three CPs were calculated (Figure 8e, Figure 9e, and Figure 10e) within a crystal radius of 10.0 Å. The calculation revealed lattice void volumes of 367.32 Å^3^, 682.64 Å^3^, and 304.17 Å^3^ in **1**, **2,** and **3**, respectively, with void areas of 75.31 Å^2^, 1032.63 Å^2^, and 847.93 Å^2^, in CPs **1**, **2,** and **3**, respectively. Hence, these calculations authenticated the microporous nature of these CPs. Furthermore, the calculated void area and volume indicate that DTZ/RhB molecules in **2** could be best absorbed and undergo decomposition. This might be the reason for the superior performance of CP **2** as a photocatalyst for the decomposition of DTZ/RhB pollutants. Furthermore, another reason for the better performance of **2** is the presence of the co-ligand 1,10-phenanthroline, which can attune the electron communication of this CP in an opposite manner than the other two CPs devoid of any co-ligand.

Hence, the plausible photocatalysis takes place as follows: On irradiation, the electrons (e^−^) in these catalysts get photo-excited from the valence band (VB) → conduction band (CB) with concomitant generation of an equivalent number of holes (h^+^) in VB. The superoxide radicals (O_2_^•−^) thereafter are produced by the reaction of molecular oxygen with e^−^ and also by oxidizing hydroxide ions by h^+^. These reactive oxygen species collectively decompose DTZ/RhB molecules in the aqueous medium.

## 3. Conclusions

In the presented work, three new coordination polymers comprising Ni(II), Mn(II), and Cd(II) metal centers have been synthesized using 2-(3,5-dicarboxyphenyl)-6-carboxybenzimidazole and 1,10-phenanthroline as co-ligands in a Mn(II)-based coordination polymer. These CPs were used as photocatalysts against the photodecomposition of dinitrazole (DTZ) antibiotics and rhodamine B (RhB) dye. The photocatalytic experiments indicated that amongst all three CPs, the Mn(II)-based CP displayed the best photocatalytic performance and degraded 91.1% of DTZ and 95.0% of RhB within 45 min. The plausible photocatalytic degradation mechanism has been proposed with the help of trapping experiments, which suggested that the presence of superoxide radicals play a pivotal role in the decomposition of Mn(II)-CP and assisted decomposition of DTZ and RhB. In addition, better performance of Mn(II)-CP was accredited to the presence of the co-ligand 1,10-phenanthroline, which can attune the electron communication of this CP in an opposite manner than the other two CPs devoid of any co-ligand. Hence, it can be concluded that the mixed ligand strategy engenders CPs having attuned electron communication with better antibiotic/dye adsorption and radical generation properties. Such CPs could be exploited as photocatalysts for the safe and sustainable degradation of aromatic dyes and antibiotics in an aqueous medium.

## 4. Experimental

### 4.1. Materials and Method

All chemicals were of analytical grade and used without additional purification. PXRD measurements were performed on a Bruker ADVANCE X-ray diffractometer employing Cu-Kα radiation (λ = 1.5418 Å,) (Bruker, Billerica, MA, USA), while FT-IR spectral data in KBr discs were collected on a Nicolet Impact 750 FTIR spectrometer.

### 4.2. Syntheses

#### 4.2.1. Synthesis of **1**

A mixture of Ni(NO_3_)_2_·6H_2_O (0.0291 g, 0.1 mmol), 2-(3,5-dicarboxyphenyl)-6-carboxybenzimidazole (0.0326 g, 0.1 mmol), and deionized water (10 mL) was stirred for 30 min in air. The resulting solution was placed in a Teflon-lined stainless-steel vessel (25 mL) and heated to 180 °C for 72 h. The solution was then cooled to room temperature at a rate of 5 °C h^−1^, and green block crystals of **1** were obtained. (Yield 45% based on Ni). Anal. (%) calcd for **1**: C, 42.36; H, 3.49; N, 6.18. Found: C, 42.33; H, 3.45; N, 6.20. IR(cm^−1^): 3350 m, 2361 w, 1680 v, 1500 v, 1400 v, 1296 w, 1375 s, 1235 s, 841 m, 757 m, 709 w.

#### 4.2.2. Synthesis of **2**

A mixture of Mn(NO_3_)_2_·4H_2_O (0.0251 g, 0.1 mmol), 2-(3,5-dicarboxyphenyl)-6-carboxybenzimidazole (0.0326 g, 0.1 mmol), 1,10-phenanthroline (0.0180 g, 0.1 mmol), and deionized water (10 mL) was stirred for 30 min in air. The resulting solution was placed in a Teflon-lined stainless-steel vessel (25 mL) and heated to 180 °C for 72 h, and pink block crystals of **2** were obtained. (Yield 55% based on Mn). Anal. (%) calcd for **2**: C, 54.12; H, 3.00; N, 9.02. Found: C, 54.20; H, 2.96; N, 9.08. IR(cm^−1^): 3432 m, 3077 w, 1625 v, 1565 v, 1513 s, 1384 s, 1165 m, 1090 s, 787 w, 721 w.

#### 4.2.3. Synthesis of **3**

A mixture of Cd(ClO_4_)_2_·6H_2_O (0.0420 g, 0.1 mmol), 2-(3,5-dicarboxyphenyl)-6-carboxybenzimidazole (0.0326 g, 0.1 mmol), and deionized water (10 mL) was stirred for 30 min in air. The resulting solution was placed in a Teflon-lined stainless-steel vessel (25 mL) and heated to 160 °C for 72 h, and yellow block crystals of **3** were obtained. (Yield 55% based on Cd). Anal. (%) calcd for **3**: C, 40.66; H, 2.56; N, 5.93. Found: C, 40.70; H, 2.60; N, 5.89. IR (cm^−1^): 3420 m, 1621 v, 1568 v, 1392 s, 1110 s, 787 m, 710 s.

### 4.3. X-ray Crystallography

The single crystal X-ray diffraction data were recorded on a Bruker SMART APEX diffractometer (Billerica, MA, USA) using monochromatic Mo-Kα radiation (λ = 0.71073 Å) using the ɷ-scan method. The structure solutions were performed employing a direct method (SHLEXS-2014), and refinement was executed using a full-matrix least-square procedure based on *F*^2^ (SHELXL-2014) [96]. Using a riding model, all hydrogen atoms were geometrically created and refined isotropically, while anisotropic displacement parameters were used for the refinement of all non-hydrogen atoms. The crystallographic information and relevant geometrical parameters for **1**–**3** are shown in Appendix A. CCDC numbers: 2289725 (**1**); 2289726 (**2**); and 2289727 (**3**)

### 4.4. Photocatalytic Method

For assessment of the photocatalytic properties, the reaction mixture was prepared by adding finely powdered CP (25 mg) to 50 mL aqueous solution of DTZ/RhB (20 μmol/L). This mixture was agitated in the dark for 30 min to attain adsorption–desorption equilibrium. Thereafter, the reaction mixture was placed in a UV-400 type photochemical reactor equipped with a 400 W mercury lamp (λ = 365 nm) (Shanghai Libi Vacuum Technology Co., Shanghai, China) for photocatalysis. During photocatalysis, batches of 5.0 mL were isolated after every 5 min and the DTZ/RhB solution separated from CP by centrifugation, and then the UV-Vis spectra were recorded. Furthermore, control experiments were executed under similar reaction conditions but without incorporation of photocatalysts in the solutions of DTZ/RhB.

### 4.5. Computational Details

Molecular Hirshfeld surfaces in the crystal structures of all three CPs were constructed by using the procedure mentioned previously [93,94,95].

## Figures and Tables

**Figure 1 molecules-28-07318-f001:**
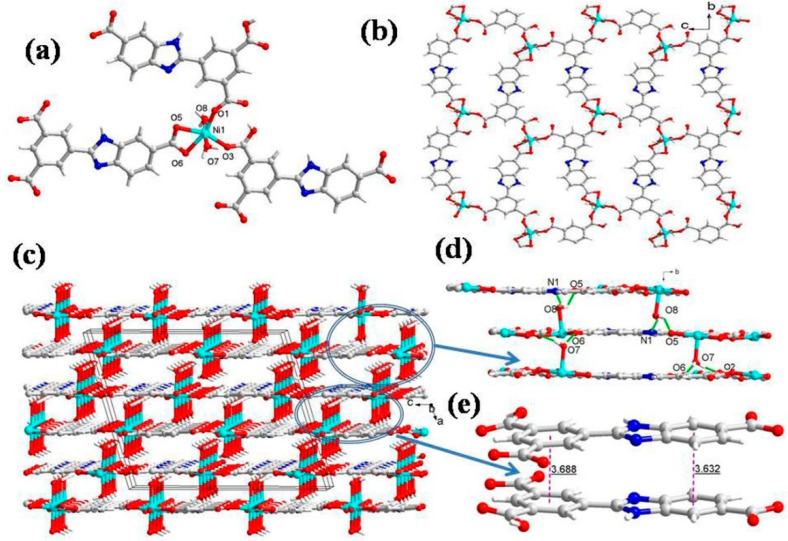
(**a**) Asymmetric unit displaying coordination environment around Ni(II) in CP **1**; (**b**) 2D layer viewed along the *bc* plane; (**c**) 3D supramolecular network; (**d**) the different hydrogen bonding interactions (**e**) π···π stacking within 3D supramolecular net for CP **1**.

**Figure 2 molecules-28-07318-f002:**
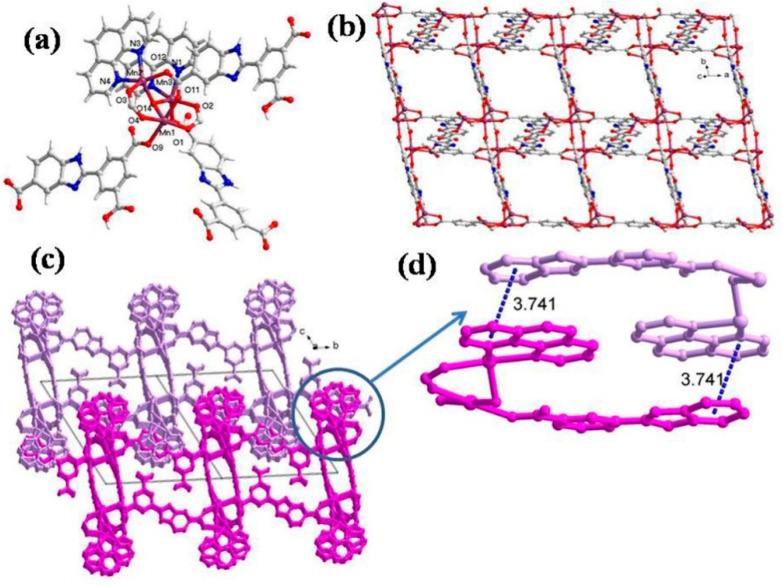
(**a**) Asymmetric unit displaying coordination environment around Mn(II) in CP **2**; (**b**) 2D layer viewed along the *ac* plane; (**c**) 3D supramolecular network; (**d**) π···π stacking mode.

**Figure 3 molecules-28-07318-f003:**
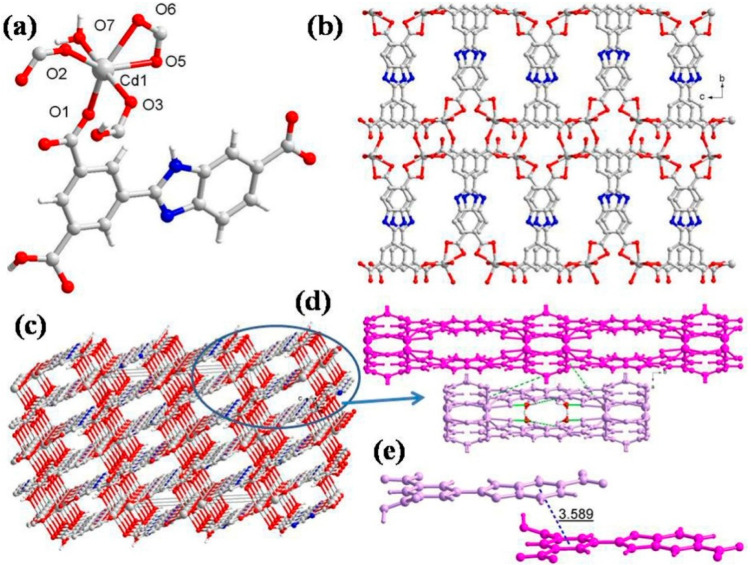
(**a**) Asymmetric unit displaying coordination environment around Cd(II) in CP **3**; (**b**) 2D layer viewed along the *bc* plane; (**c**) 3D supramolecular network; (**d**) different hydrogen-bonding interactions; (**e**) π···π stacking within 3D supramolecular net for CP **3**.

**Figure 4 molecules-28-07318-f004:**
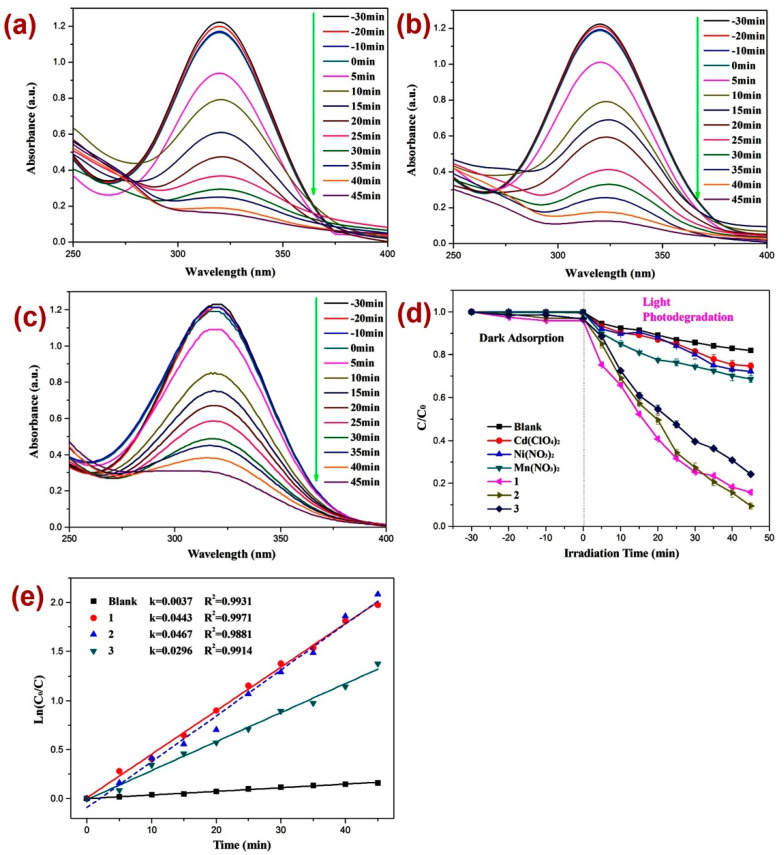
Photodegradation of DTZ in the presence of photocatalysts (**a**) 1; (**b**) 2; (**c**) 3; (**d**) graph representing the plot of *C/C*_0_ vs. irradiation time, where C/C_0_ represents the concentration ratio of DTZ in the presence of metal salts and their corresponding CPs; (**e**) results of kinetic studies displaying pseudo-first-order kinetics.

**Figure 5 molecules-28-07318-f005:**
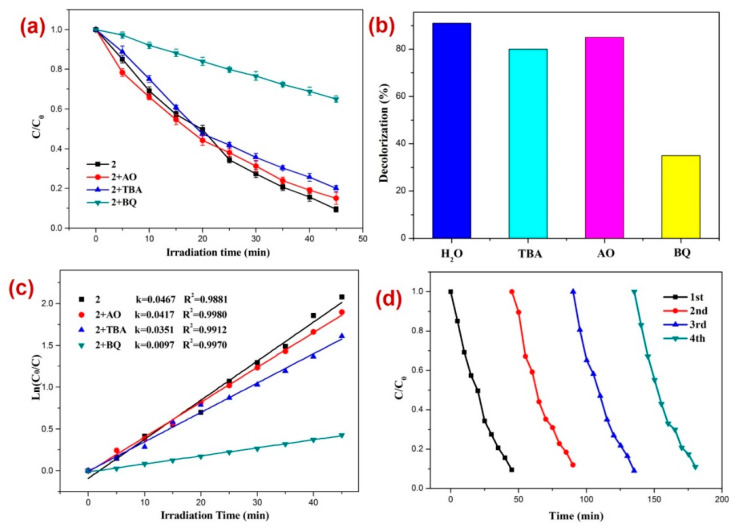
Results of the radical scavenging experiments displaying (**a**) the plot of C/C_0_ vs. irradiation time for the decomposition of DTZ in the presence of photocatalyst 2 and different radical scavengers; (**b**) bar graphs presenting percentage decolorization of DTZ in the presence of different radical scavengers and **2**; (**c**) results of the kinetic studies; and (**d**) recycle experiments.

**Figure 6 molecules-28-07318-f006:**
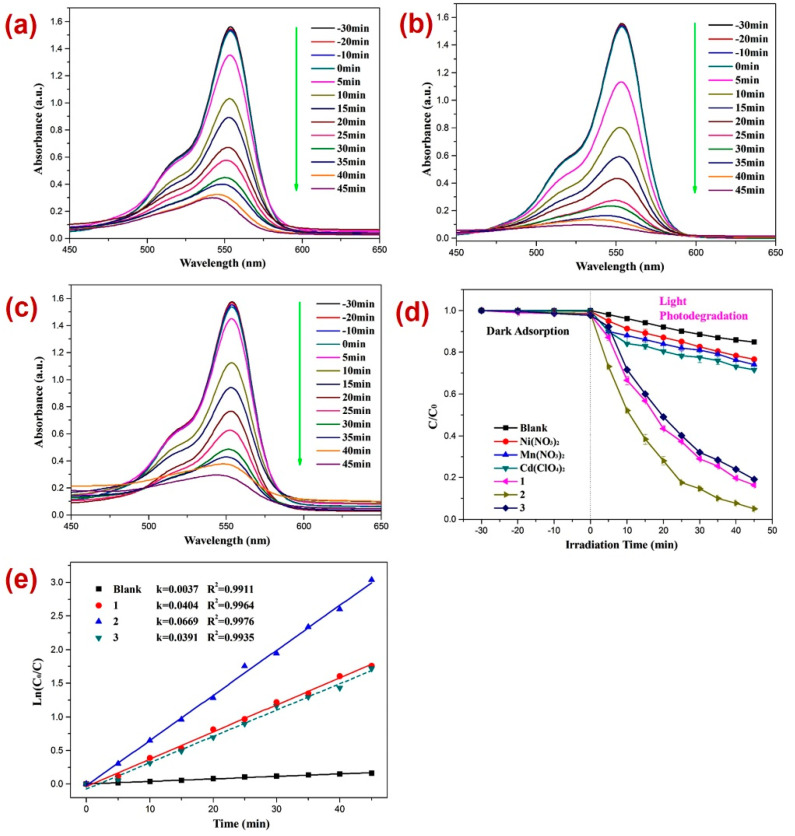
Photodegradation of RhB in the presence of photocatalysts (**a**) 1; (**b**) 2; (**c**) 3; (**d**) graph representing the plot of *C/C*_0_ vs. irradiation time, where C/C_0_ represents the concentration ratio of RhB in the presence of metal salts and their corresponding CPs; (**e**) results of kinetic studies displaying pseudo-first-order kinetics.

**Figure 7 molecules-28-07318-f007:**
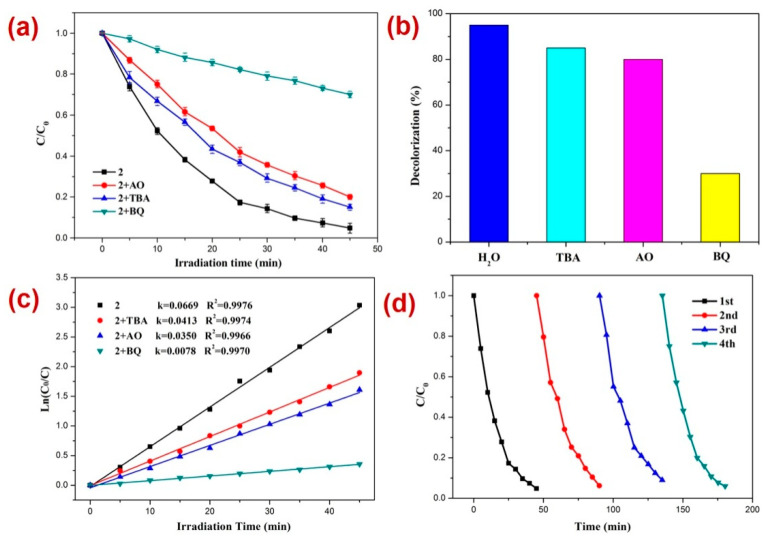
Results of the radical scavenging experiments displaying (**a**) the plot of C/C_0_ vs. irradiation time for the decomposition of RhB in the presence of photocatalyst 2 and different radical scavengers; (**b**) bar graphs presenting percentage decolorization of RhB in the presence of different radical scavengers and **2**; (**c**) results of the kinetic studies; and (**d**) recycle experiments.

**Figure 8 molecules-28-07318-f008:**
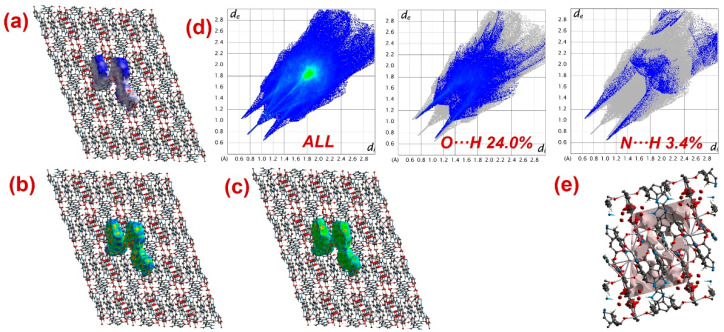
Results of the Hirshfeld surface analysis for **1** presented as (**a**) *d*_norm_; (**b**) shape index; (**c**) curvedness; (**d**) full and partial fingerprint plots; (**e**) void volume calculation results.

**Figure 9 molecules-28-07318-f009:**
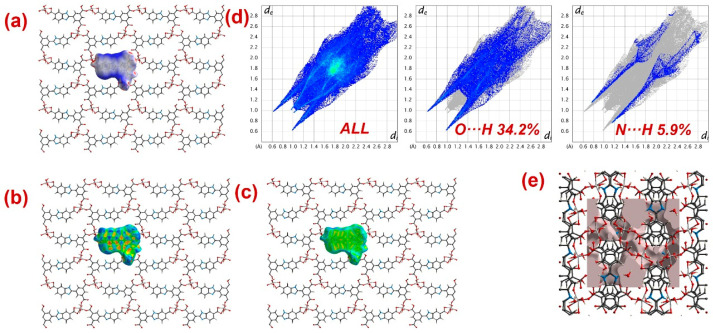
Results of the Hirshfeld surface analysis for **2** presented as (**a**) *d*_norm_; (**b**) shape index; (**c**) curvedness; (**d**) full and partial fingerprint plots; (**e**) void volume calculation results.

**Figure 10 molecules-28-07318-f010:**
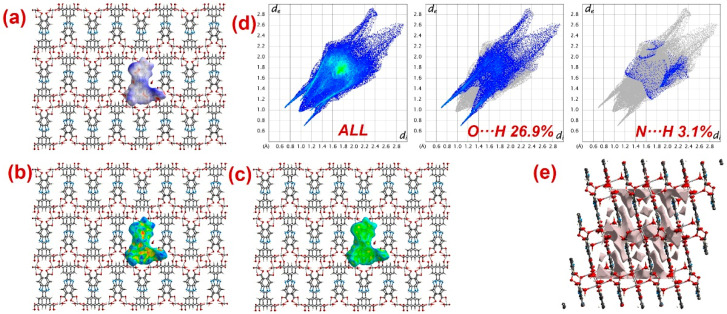
Results of the Hirshfeld surface analysis for **3** presented as (**a**) *d*_norm_; (**b**) shape index; (**c**) curvedness; (**d**) full and partial fingerprint plots; (**e**) void volume calculation results.

**Table 1 molecules-28-07318-t001:** The different rate constants (k) obtained by fitting of the pseudo-first-order kinetics curves.

Material	DTZ		RhB	
	*k* (min^−1^)	*R* ^2^	*k* (min^−1^)	*R* ^2^
**Blank**	0.0037	0.9931	0.0037	0.9911
**1**	0.0443	0.9971	0.0404	0.9964
**2**	0.0467	0.9881	0.0669	0.9976
**3**	0.0296	0.9914	0.0391	0.9935
**2 + AO**	0.0417	0.9980	0.0350	0.9966
**2 + TBA**	0.0351	0.9912	0.0413	0.9974
**2 + BQ**	0.0097	0.9970	0.0078	0.9970

## Data Availability

Not applicable.

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
