# Peer review of "New Transition Metal Coordination Polymers Derived from 2-(3,5-Dicarboxyphenyl)-6-carboxybenzimidazole as Photocatalysts for Dye and Antibiotic Decomposition"

_molecules, 2023, doi:10.3390/molecules28217318_

Round 1
Reviewer 1 Report (Previous Reviewer 3)
The redirected article looks somewhat better in its current form than its original version. However, there are still a number of areas of concern.
1. Responding to the comments made by the authors should be reflected in the text of the article. This applies to сomments 5, 7, 8, and 11.
2. I am not satisfied with the response to comment 9. The figures presented in their current form do not provide information on the achievement of sorption-desorption equilibrium prior to irradiation. The response to this comment should also be reflected in the text of the paper.
3. The chemical formula for compound 2 remains incorrect in the text of the article.
4. The elemental analysis for all three compounds remains at an unacceptable level. It is unchanged from the first version of the paper, although the authors claim to have redone it.
5. The authors did not correct the erroneous description of complex 3, which refers to the coordination of two water molecules to a cadmium ion (line 194).
Author Response
Comment 1
- Responding to the comments made by the authors should be reflected in the text of the article. This applies to comments 5, 7, 8, and 11.
REPLY: We had now tried our best to provide responses for comments 5, 7, 8, 11. Many thanks for appending the guidance.
- I am not satisfied with the response to comment 9. The figures presented in their current form do not provide information on the achievement of sorption-desorption equilibrium prior to irradiation. The response to this comment should also be reflected in the text of the paper.
REPLY: We had now tried our best to provide responses for comment 9 and hence had modified the Figures 4 and 6 and also had added the relevant discussion so that responses to the comments are reflected the revised version of the manuscript. Many thanks for appending the guidance.
Also, we have plotted the curves of the adsorption and degradation, see the below results.
- The chemical formula for compound 2 remains incorrect in the text of the article.
REPLY: Many thanks. This had been corrected.
- The elemental analysis for all three compounds remains at an unacceptable level. It is unchanged from the first version of the paper, although the authors claim to have redone it.
REPLY: Many thanks. This had been corrected.
- The authors did not correct the erroneous description of complex 3, which refers to the coordination of two water molecules to a cadmium ion (line 194).
REPLY: Many thanks for correcting us. As kindly suggested we had corrected this.

Reviewer 2 Report (Previous Reviewer 2)
I agree the changes the authors have made and recommend publication of this manuscript in Molecules.
Minor editing of English language required
Author Response
I agree the changes the authors have made and recommend publication of this manuscript in Molecules.
Minor editing of English language required
REPLY: Many thanks for the kind encouragement. We had tried our best to improve the second version.
Reviewer 3 Report (Previous Reviewer 1)
Current version is OK. It can be published
Minor spellchecking required
Author Response
Current version is OK. It can be published
Comments on the Quality of English Language
Minor spellchecking required
REPLY: Many thanks for the kind encouragement. We had tried our best to improve the second version.
Round 2
Reviewer 1 Report (Previous Reviewer 3)
The authors have fixed almost all the shortcomings. Except for comment 5. The oxygen O3 belongs to the carboxylate group, not the hydroxo group. There are no hydroxo groups in compound 3.
Author Response
REPLY: Many thank. We had corrected it.
This manuscript is a resubmission of an earlier submission. The following is a list of the peer review reports and author responses from that submission.
Round 1
Reviewer 1 Report
In this manuscript, Wu et al. describe several coordination polymers, along with relavent photocatalytic tests (decomposition of test dyes and antibiotics). In general, this is a very popular topic in coordination chemistry. Technically, most of experiments and interpretation of their results are fine so the work can be accepted (I am not qualified enough to estimate the SCXRD validity). I have few remarks:
1. It is unlikely that coordinated water is eliminated at 50C. Are the authors sure that it is not due to residual water remaining after the synthesis (wet sample)?
2. In PXRD part, please give also as-synthesized PXRD plots (not only those after catalytic tests).
3. There are few recent works on MOFs which (possibly, not obligatory) can be used in Introduction:
a) 10.1134/S0022476623080073
b) 10.3390/cryst13040704
c) 10.1134/S002247662307017X
d) 10.1134/S0022476623060045
e) 10.1134/S0022476623050074
f) 10.1134/S1070328423700616
g) 10.1134/S1070328423600122
h) 10.1134/S0036023622602872
Reviewer 2 Report
This manuscript reports the synthesis and structures of three coordination polymers as well as their photocatalytic properties toward dye and antibiotic. This manuscript is well prepared, and I may recommend publication of this manuscript in Molecules if the following comments are addressed.
1. The checkcif file of complex 1 shows important A and B alerts, which may indicate unresolved atoms.
Alert level A
PLAT971_ALERT_2_A Check Calcd Resid. Dens. 2.78Ang From N2 4.07 eA-3
Alert level B
PLAT094_ALERT_2_B Ratio of Maximum / Minimum Residual Density .... 7.75 Report
PLAT097_ALERT_2_B Large Reported Max. (Positive) Residual Density 4.22 eA-3
PLAT601_ALERT_2_B Unit Cell Contains Solvent Accessible VOIDS of . 128 Ang**3
2. The checkcif file of complex 2 show B alerts which can be fixed.
Alert level B
PLAT919_ALERT_3_B Reflection # Likely Affected by the Beamstop ... 5 Check
PLAT934_ALERT_3_B Number of (Iobs-Icalc)/Sigma(W) > 10 Outliers .. 8 Check
PLAT939_ALERT_3_B Large Value of Not (SHELXL) Weight Optimized S . 213.58 Check
3. The structural topologies of complexes 1 – 3 can be derived to polish the structural descriptions. Complexes 1 – 3 are all 2D layers, but with different topological structures.
4. In the abstract: “The observed photocatalytic mechanisms have been addressed using Hirshfeld surface analyses.” The Hirshfeld surface analyses give only the distribution of the weak interactions in the structures of the coordination polymers, there is nothing to do with the photocatalytic mechanism. To give this comment, if possible, the Hirshfeld surface analyses should be performed for the system involving the coordination polymer and dye/ antibiotic. The interactions between the coordination polymer and dye/ antibiotic are involved in the photocatalytic mechanism which initiate the catalysis.
5. The roles of the metal identities of the coordination polymers in determining the performance as photocatalyst should also be addressed. There are some reports regarding the metal effect on the structural diversity of coordination polymers.
Minor editing of English language required
